# Decorative Chromium Coatings on Polycarbonate Substrate for the Automotive Industry

**DOI:** 10.3390/ma16062315

**Published:** 2023-03-14

**Authors:** Filipa Ponte, Pooja Sharma, Nuno Miguel Figueiredo, Jorge Ferreira, Sandra Carvalho

**Affiliations:** 1CEMMPRE, Department of Mechanical Engineering, University of Coimbra, Rua Luís Reis Santos, 3030-788 Coimbra, Portugal; poojamnit2014@gmail.com (P.S.); nuno.figueiredo@dem.uc.pt (N.M.F.); sandra.carvalho@dem.uc.pt (S.C.); 2CFUM-UP, Centro de Física das Universidades do Minho e do Porto, University of Minho, Campus of Azurém, 4800-058 Guimarães, Portugal; 3Engineering Department, KLC—Technical Plastics, 2430-021 Marinha Grande, Portugal; jorge.ferreira@klc.pt; 4IPN—LED & MAT—Instituto Pedro Nunes, Rua Pedro Nunes, 3030-199 Coimbra, Portugal

**Keywords:** polycarbonate, chromium coating, plasma etching, sputtering, automotive industry

## Abstract

Metal-coated plastic parts are replacing traditional metallic materials in the automotive industry. Sputtering is an alternative technology that is more environmentally friendly than electrolytic coatings. Most metalized plastic parts are coated with a thin metal layer (~100–200 nm). In this work, the challenge is to achieve thicker films without cracking or without other defects, such as pinholes or pores. Chromium coatings with different thicknesses were deposited onto two different substrates, polycarbonate with and without a base coat, using dc magnetron sputtering in an atmosphere of Ar. Firstly, in order to improve the coating adhesion on the polymer surface, a plasma etching treatment was applied. The coatings were characterized for a wide thickness range from 800 nm to 1600 nm. As the thickness of the coatings increased, there was an increase in the specular reflectivity and roughness of the coatings and changes in morphology due to the columnar growth of the film and a progressive increase in thermal stresses. Furthermore, a decrease in the hardness and the number of pinholes was noticed. The maximum thickness achieved without forming buckling defects was 1400 nm. The tape tests confirmed that every deposited coating showed a good interface adhesion to both polymers.

## 1. Introduction

The automotive industry was a major driver for the mass production of thermoplastics, due to their widespread use from the 1950s onwards. The use of thermoplastics, such as ABS, polyamides, and polycarbonates, in the industry became increasingly common to replace metal components. The reasons that lead to the constant growth of plastic components in the automotive industry are their low-weight, which allows a more energy-efficient system, corrosion resistance, great freedom of design, and the low production cost, which are characteristic of polymers [1,2,3]. Most of the used plastic components have been manufactured by injection molding and further metalized by other methods, such as electrochemical plating, PVD (Physical Vapor Deposition), a family of thermal spray methods, and a series of polymer–metal direct bonding methods (such as adhesive bonding, injection overmolding, and fusion joining techniques) [4]. In this way, it was possible to combine the beneficial characteristics of plastics with those of the metals that coat them, i.e., maintaining a shiny metallic finish and a high reflectivity and conductivity [3]. The chrome plating process, in particular, can provide greater protection against wear and corrosion while giving the parts a metallic decorative appearance [5]. In this process, however, coatings have been mostly produced from a chemical bath containing hexavalent chromium ions. Contact with this ion has proven to be dangerous for the environment and human health [5]. Over the years, it has been found that there are several health problems associated with exposure to hexavalent chromium [6], among which are eye irritation, respiratory problems (asthma), lung cancer, nasal irritation, and other nasal damage, as well as skin irritation, ulcers, or allergy [7,8,9]. On June 2007, the European Union’s REACH regulation (Registration, Evaluation, Authorization, and Restriction of Chemicals) came into force. This regulation has the power to restrict or ban the use of certain substances (depending on their risks) and to replace fuel chemicals with less dangerous ones, promoting a constant sustainable development of the industry [5,10,11,12,13]. That said, on 21 September 2017, the use of chromium VI in coatings was banned in the European Union, forcing companies to find feasible alternatives to the chrome plating process, which are at the same time more environmentally friendly [5].

The mainstream intention is to replace processes that use chemical baths with other high-performance dry coating techniques. There are several possible deposition technologies, such as PVD, CVD (Chemical Vapor Deposition), or thermal spray [5,14]. The goal is that these processes are equally effective, allowing the deposition of metallic coatings, in this case, chromium, for various possible applications such as decorative protective surfaces or in tools [5,14]. The PVD processes, namely sputtering, are excellent options for cases where one intends to improve wear and corrosion resistance, in applications such as tools, decorative parts, molds, and others. The sputtering process has several advantages, such as producing coatings with good adhesion, homogeneity and controlled morphology, as well as the possibility of making multilayer coatings and using a wide range of materials [15]. 

In this study, chromium coatings with incremental thicknesses were deposited by magnetron sputtering onto polycarbonate (PC) substrates. PC is being increasingly used and adapted to various industries, including the automotive industry, and can be used for both technical and optical applications because of its interesting properties, such as tunable optical transmittance, excellent thermal and flame resistance, high impact withstanding strength, and high stability under different environmental conditions [3,16]. Since the direct application of a metallic coating on a polymeric part may present some challenges and compromise the durability of the final part, one of the main common solutions is to apply a base-coat layer after obtaining the polymer to (i) level the surface, allowing a flatter deposition of the metallic layer and (ii) improve the adhesion properties [17]. Thus, one of the main challenges is to achieve metallic-looking coatings on polycarbonate surfaces without the need for applying any base coat. Another challenge lies in depositing thick Cr films on polymers. From the literature, Cr coatings were deposited on polymers in a thickness range between ~100 nm and ~600 nm [3,15,18]. Thicker films (>1000 nm) are often desired, especially for decorative applications, since they show higher impact abrasion resistance—however, there is a trade-off, since increasing the thickness of the metallic films may increase the propensity for coating delamination [19]. In the case of functional coatings, the occurrence of cracking or delamination due to excessive residual stress levels can dramatically affect the performance, reliability, and durability of material components [20]. In this study, chromium films with higher thicknesses than those reported in the literature (in a range between 800 and 1600 nm) were tested over two different types of PC substrates, with and without an industrial-grade base coat. The two main objectives were (i) to maximize film thickness while (ii) obtaining high-quality sputtered chromium films (i.e., having good optical properties, improved adhesion, reduced density of pinhole defects, and minimal cracking/delamination).

## 2. Materials and Methods

Cr films were deposited by magnetron sputtering, in a custom-made sputtering chamber (Figure 1), on two types of polycarbonates, PCB (transparent PC with transparent base coat, based on acrylate resins) and PCW (white opaque PC without any base coat), and silicon (100) with an area of 18 × 18 mm^2^. To estimate the temperature of the chamber, a Testoterm™ temperature measuring tape was tapped into the substrate holder, recording the temperature (by changing color) in a range between 37 °C and 65 °C. 

All the substrates were cleaned in an ultrasonic bath to remove dirt impurities and/or oils and improve the adhesion of the coatings. The PC substrates were cleaned using isopropanol for 10 min, whereas silicon substrates were cleaned using two liquids, first acetone and then ethanol, for 10 min each. 

All substrates were mounted on the substrate holder and, after loading the substrate holder into the deposition chamber, the pumping was performed until reaching a base pressure of ~8 × 10^−4^ Pa. Before each deposition, the substrates were plasma-treated for 2.5 min in an argon atmosphere (37.5 sccm of Ar flow), with a pulsed dc bias voltage of -300 V applied to the substrate holder and the frequency and off-time 250 kHz and 1.6 μs, respectively. The pressure during the plasma treatment was maintained at ~1 Pa. Simultaneously, the Cr target (99.9% purity) was connected to a dc power supply, and a power of 400 W was applied in order to remove any contaminants from its surface. During the plasma treatment process, a stainless steel shutter was placed in between the target and the substrate holder to avoid cross-contamination.

After plasma treatment, Cr coatings of different thicknesses (800 nm, 1400 nm, and 1600 nm) were deposited on the substrates using 1200 W of target power. The substrate holder was kept at a floating potential and at a constant rotation speed of 18 rpm. A deposition rate of ~0.74 nm/s was obtained for these deposition conditions.

The surface morphology of the coatings was studied by Scanning Electron Microscopy (SEM) in a Hitachi, model Su3800, using both planar (on polymers) and cross-sectional views (on Si—sample preparation consisted of scratching the Si substrates in a corner and then breaking them carefully by propagation, exposing the new surface without any type of contact with any material or surface). The topographic information of the coatings, such as surface feature size and surface roughness, was obtained by Atomic Force Microscopy (AFM) in a Bruker Innova, in tapping mode (using a silicon nitride tip). 

To analyze the structure of the chromium coatings deposited on polycarbonate substrates, the samples were evaluated by X-ray Diffraction (XRD) in two different configurations. Bragg–Brentano scans were performed in a Rigaku SmartLab diffractometer (Rigaku Corporation, Tokyo, Japan), using CuKα radiation, operating at a tube voltage of 40 kV and a tube current of 50 mA. The 2θ range varied between 35 and 145°, with a step width of 0.02° and a speed duration of 10°/min. Grazing Incidence scans were performed in a Philips X’Pert diffractometer (PANalytical, Almelo, The Netherlands) with CoKα (λ = 1.789010 Å) radiation, operating at 40 kV and 35 mA, and using an incidence angle of 2°. The tests were performed in parallel beam geometry, with a 2θ range set from 20° to 120°, a step size of 0.025°, and an exposure time of 1 s per step. According to each equipment and reflection, the measured (experimental) peak width was corrected by subtracting the corresponding instrumental peak width. For the θ/2θ configuration scans, it was possible to analyze the degree of out-of-plane preferred crystallographic orientation or texture by determining the texture coefficients [21]. The Scherrer formula [22] was used to estimate the crystallite size from the diffraction peaks, using Pseudo-Voigt profiles to fit each peak. The residual stresses of the films were calculated according to Stoney’s Equation [23], by analyzing the curvature profile of thin films deposited over flat silicon substrates. The curvature profiles were determined by profilometry using a Mahr Perthometer S4P (Providence, Rhode Island) with a mechanical probe head (Perth RFHTB-50). The thickness of the coatings was also determined in the same equipment by measuring the step between an uncoated and a coated area of the substrate.

The adhesion of the coatings to the polymeric substrates was tested using the Tape-test, according to ISO 2409. Following the standard guidelines, two sets of six cuts were made, perpendicular to each other, with 2 mm of spacing between them, forming a grid. After this step, a standardized tape (Tesa^®^ 4657) was applied to the cross-section for five minutes and then removed with constant force at an angle as close as possible to 60°. As described in the standard, according to the amount of coating removed, specifically the number of squares removed, a rating from zero to five is assigned as an indication of whether it is adhering properly to the substrate. A rating of zero corresponds to a coating with perfect adhesion where the edges of the cuts are completely smooth and none of the grid squares have peeled off, while a rating of five is equivalent to a coating that has peeled off along the edges of the cuts in large strips and/or squares that have partially or completely peeled off to an extent greater than 65% of the tested area [24].

The reflectivity of the coated substrates was measured in a Gretagmacbeth ColorEye^®^ XTH spectrophotometer (Regensdorf, Switzerland), in a wavelength range between 360 nm and 750 nm. This equipment allows the analysis of both specular and diffuse reflection components. In this study, only the diffuse reflection component was considered since it can be more easily related to the different features/defects of the coatings (e.g., surface roughness, microstructure, cracks, pinholes). The color coordinates of the coatings were also measured in the CIE-L*a*b* color space, while considering both specular and diffuse components. 

For the pinhole analysis, a ZEISS^®^ optical microscope was used with retro-illumination at 5× objective. To allow pinhole quantification, 9 pictures were taken per sample, in a matrix grid of 3 × 3 pictures.

The hardness (*H*) and reduced Young’s modulus (*Er*) of the films deposited on Si substrates were measured by nanoindentation in a NanoTest equipment, with a Berkovich diamond pyramid indenter. A total of 25 indentations were performed for each sample, where the hardness resulted in their calculated average. A maximum load of 3 mN was set with a loading/unloading rate of 0.1 mN/s. After the maximum load value was reached, it was applied for 30 s.

## 3. Results and Discussion

Two main sections are presented in this paper: (i) Structure, Microstructure, and Surface Morphology of the films and (ii) their Functional Properties.

### 3.1. Structure, Microstructure, and Surface Morphology

For the successive use of these coatings in the automobile industry, it is a prime factor that the coatings must be free of defects and at the same time possess high thickness. As the thickness increases, the films have greater abrasive resistance, but there is also a higher chance of cracking the coating [19]. Figure 2 shows the plain-view SEM images of polymeric substrates coated with the following thicknesses: 800 nm, 1400 nm, and 1600 nm. To complement this analysis, topographic information was accessed by AFM, as shown in Figure 3 and Figure 4.

From Figure 2, the existence of a transition region is clear: Cr thin films without any major defects can be deposited for thicknesses up to 1400 nm, after which the films begin to form pronounced buckling for both substrates (1600 nm—Figure 2e,f). Although 2D SEM micrographs alone are not able to provide any reliable information on the height profile of the surface defects, the 3D AFM micrographs in Figure 4 confirmed the presence of buckling-driven hillocks over the sample with 1600 nm. No cracking was observed. For lower coating thicknesses of 800 nm and 1400 nm, no buckling phenomena take place.

Depending upon the coating/substrate modulus of elasticity and interfacial adhesion energy and defects, the residual compressive stress may result in buckling-driven hillocks or delamination with high tensile stress at the crest of the delaminated buckle. Above a certain critical in-plane stress, this phenomenon will cause the development of cracks along the crest [25]. According to Bradley et al. [26], the residual stress, which depends strongly on the deposition conditions, is composed of three components: intrinsic, thermal, and extrinsic. The intrinsic stress reflects the unfavorable interactions between the substrate and the coating material and the resulting flaws that develop during the deposition. The thermal stress within a thin film is due to the difference in coefficients of thermal expansion of the two materials in contact. The extrinsic stress is related to the impact of any external factors on the final coated material. The occurring stress often results from different thermal expansion coefficients of the film and substrate [27]. In the present case, polycarbonate and chromium have very different thermal expansion coefficients (~66 × 10^−6^/K and ~6 × 10^−6^/K, respectively [28]), leading to the formation of compressive stresses in the films during the cool-down step. Generally, the thicker the coating is, the longer is the deposition time and, consequently, an increased surface bombardment takes place leading to higher film/substrate temperature at the end of the deposition. During cooling, in the present case, the PC substrates will contract more than the Cr films, creating compressive stresses that are high enough to generate buckling in the films at 1600 nm. Appendix A shows the evolution of substrate temperature with the variation of thickness. Although the maximum temperature observed in the substrate holder, at 1600 nm, was 55 °C, this was high enough to induce compressive stresses in the films during cooling. Just as a side note, the maximum observed substrate temperature value (55 °C) was much lower than the glass transition temperatures of these polymers (~150 °C).

From the AFM micrographs in Figure 3, it was possible to calculate the surface feature size and root means square (RMS) roughness (Table 1). Both surface feature size and RMS roughness increase with the coating thickness on PCB and PCW. The surface feature size increase with increasing thickness is naturally related to the expected columnar growth for these coatings. With increasing thickness, it is also noticeable that there are more individual columns with different heights and that the columns tend to become more elongated along the direction perpendicular to the substrate rotation (X axis in Figure 3a). In addition, as it is evidenced in Figure 3e,f, above a certain film thickness the columns begin to elongate along the direction perpendicular to the substrate rotation. Since the substrate holder is rotating within the deposition chamber, the angle of incidence of the atoms varies more pronouncedly along the direction of rotation (Y axis in Figure 3a), leading to obliquely incident atoms being deposited preferentially on hills of the surface, causing an atomic shadowing effect. The shadow effect causes growth competition between individual columns that contributes to the elongation of the columns in the direction perpendicular to substrate rotation and to the increase in the surface roughness of the films [21,29,30,31].

Figure 5 shows the XRD results obtained for the Cr films deposited on PCB (Figure 5a) and PCW (Figure 5b), for all film thicknesses. All XRD patterns exhibit peaks centered at ~44.4°, 81.7°, and ~135.4°, corresponding to the (110), (211), and (222) planes of the chromium b.c.c structure, respectively. The crystallite size was calculated for the (222) and (110) planes and was added to the pictures. In general, a decrease in crystal size was observed with the film thickness. From the obtained diffractograms, the texture coefficient was determined, as shown in Figure 6. A texture coefficient higher than 1 indicates a preferred orientation along the corresponding direction. With the thickness increase, an increasing preferential grain growth was observed along the [111] direction and a progressive restrained grain growth along the [110] direction. According to Oliveira et al. [21], the [111] preferential orientation is characteristic of Cr films deposited at higher pressures or with higher film thicknesses and could be generally associated with the deposition of more open, porous microstructures. On the other hand, O. P. Karpenko at al. [32] also reported that the more oblique the incident flux angle and the more anisotropic the grain shape, the faster the rate of film texturization is expected to be. Thus, the degree of texturization should increase with the coating thickness, as observed in the present case.

To complement the structure analysis, another XRD configuration was tested in order to gather additional information on the crystallographic orientations that are non-parallel to the substrate surface. In the symmetric θ/2θ configuration, the scattering vector Q is always oriented at 90° to the surface of the film. The crystallographic lattice planes that give rise to an observed Bragg reflection are all oriented parallel to the substrate plane. In the grazing incidence configuration, however, Q is oriented at (90 + θ − α)° to the surface of the film (where α = 2° in the present case). Thus, reflections with distinct Bragg angles θ_hkl_ arise from lattice planes that are neither parallel to the substrate surface nor to each other. In this case, since Cokα radiation was used, it was only possible to analyze the (110) and (211) reflections. This analysis corresponded, respectively, to planes that are rotated 24° and 47 ° away from the substrates’ surface plane. Figure 7 shows the grazing incident (GI) XRD results obtained for the Cr films deposited on PCB (Figure 7a) and PCW (Figure 7b) for all film thicknesses.

In contrast with the previous XRD results, where the (110) reflections progressively decreased in intensity with the thickness increase (planes parallel to the substrate surface), the (110) reflections of planes tilted 24° away from the substrate surface increased in intensity with the film thickness increase. The (110) planes have the lowest energy in the b.c.c.-based structures and, for Cr, the (110) preferential orientation develops in the early stages of the film growth, either due to energy minimization before island coalescence or by competitive growth in the subsequent stages of film growth [21]. As the thickness of the films increases, larger column tops and higher surface roughness are expected, causing the (110) planes to become more aligned with the tilted surfaces. 

In order to relate the surface feature size calculated by AFM with the morphological features arising in this type of microstructure, the cross-sectional morphology of the films deposited over Si was analyzed by SEM, as shown in Figure 8.

From Figure 8, a columnar microstructure extending from the substrate to the top of the films can be observed, resulting from the low mobility of the Cr atoms and from the shadow effect. For the 800 nm and 1400 nm thick films (Figure 8a,b, respectively) the columns are denser and more uniform, while for the 1600 nm thick film (Figure 8c) the columns are bigger and less dense, resulting in a rougher, more heterogeneous surface. It should be noted that, in this case, the difference in thermal expansion coefficients between film and substrate (~6 × 10^−6^/K for Cr and ~3 × 10^−6^/K for Si [28]) is not enough to promote any significant residual thermal stress in the films. Therefore, any residual stresses that might arise in the films should be mostly intrinsic in nature. Residual stress analysis performed on Cr films over Si substrates confirmed that the measured values were close to zero (Appendix A shows the evolution of residual stresses of coatings deposited on Si as a function of thickness).

It is observed that the size of the column tops in Figure 8 roughly correlates with the size of the surface features observed by AFM for the case of Cr films over polymers.

### 3.2. Functional Properties

In this section, the thin film quality was evaluated, namely by testing its adhesion, color coordinates, diffuse reflectivity, and pinhole defect density. Its mechanical properties were also analyzed, namely the reduced Young’s modulus and the hardness.

#### 3.2.1. Adhesion

Adhesion is a relevant factor in determining the reliability of coatings when subject to mechanical stress. After performing the tape test (Figure 9) according to ISO-2409, no delamination was observed, and only some remnants of adhesive tape were observed on parts of the surface. Thus, following the standard classification, the adhesion of the coatings was assessed at zero for both polymeric substrates, proving that it has good adhesion, irrespective of thickness.

#### 3.2.2. Brightness and Color Coordinates

An essential requirement for a metal decorative coating to convey the silver appearance (in the automobile industry) is that it is shiny (high brightness) and greyish in color. The CIE L*a*b* color coordinates of the Cr films over PCB and PCW substrates were measured via reflectivity and the results are shown in Figure 10. Here, L* represents brightness (range from 0 to 100), where 0 corresponds to black and 100 to white. The a* coordinate represents red and green on the positive and negative axes, respectively, and the b* coordinate represents yellow on the positive axis and blue on the negative axis [3,33].

As the thickness increases, the L* parameter decreases progressively, with values close to 65 for 800 nm and 58 for 1600 nm. The a* coordinate is approximately constant and close to zero for all coatings and substrates. On the other hand, the b* parameter increases from 3.8 and 3.4 to 6.3 and 7.1 as the thickness goes from 800 nm to 1400 nm (increase in yellow hue) and then slightly decreases to 6.2 and 5.2 for 1600 nm (decrease in yellow hue) for PCB and PCW, respectively. This decrease was more pronounced for the PCW substrate.

According to the literature, the visual appearance of chromium depends on the surface morphology of the films. These results can be related to the surface feature size and roughness increase as the coating thickness increases [34,35].

Despite the slight differences in hue compared to the ideal chromium color coordinates (L* = 80.48, a* = −0.5 and b* = −0.75, [36]), all the measured color points are in a grayish region (see Appendix A), as intended for the deposition of Cr. However, for the 1600 nm thick films there is a noticeable loss of brightness. It should be noted that the brightness parameter (L*) is an important quality parameter in industrial coatings. In the literature, L* values close to 70 were found for Cr films deposited in polycarbonate [3]. Therefore, the best coatings concerning brightness and color are the ones showing no buckling (800 nm and 1400 nm).

#### 3.2.3. Diffuse Reflectivity

The total reflected light consists of two main components: specular reflection (the angle of reflection is equal to the angle of incidence) and diffuse reflection (light is scattered in different directions). Reflection of light from smooth surfaces leads to specular reflection, whereas reflection of light from rough surfaces leads to diffuse reflection [37]. In particular, it can be considered that increasing certain coating features, such as surface roughness, porosity, and/or buckling/cracking defects, increases the diffuse reflectivity component and alters the gloss of the films [38]. Therefore, for this paper, it is of greater importance to study the diffuse reflectivity component of the coatings, as is presented in Figure 11.

Figure 11 shows that as the thickness of Cr films increases, the diffuse reflectivity also increases, and it is noticeable that it is much higher for coatings with 1600 nm. These results agree well with previous general observations made in the microstructure and surface morphology section, namely that the surface roughness of the films increases progressively with the thickness for both substrates, demonstrating the correlation between these two parameters. Moreover, the coatings where the diffuse reflectivity increases dramatically—1600 nm—are the ones showing a pronounced buckling of the film due to high compressive stresses, as discussed before. These hillock-rich regions dramatically increase the light scattering at random reflection angles. Therefore, both optical and microstructural results agree well.

It is important to note that, although the coating with 1400 nm has slightly higher diffuse reflectivity values than the 800 nm one, in both depositions these values can be considered low, especially when compared to the 1600 nm film.

#### 3.2.4. Pinhole Defect Density

The presence of surface defects in the substrate, the roughness of the substrate itself, the thickness of the films, or even the chemical/physical activation process of the surface can affect film growth, so the film may not be deposited homogeneously. Therefore, the formation of defects associated with thin film growth occurs, such as pinholes, pores, and other discontinuities in the coatings [39]. These factors can affect the coalescence of the grains during the deposition of thin films, and lead to the creation of voids between them and the substrate [40]. Other factors that may also be associated with the formation of pinholes are related to the presence of particles, such as dust, on the substrate surface, poor cleaning of the substrate, or improper handling before and after deposition that can generate the appearance of defects and scratches [39,41].

Pinholes, as the name implies, are small “holes” in the surface where there is no coating and the existence of a high pinhole defect density in thin metallic films can decrease the overall quality of the decorative coating, particularly when it is used in applications that require retro-illumination, since light travels unimpeded through this hole [42], such as in logos, buttons, or other automotive system control commands. Therefore, this is an important characterization step to apply on the transparent substrates, as it is the case with the PCB substrate (Figure 12).

Figure 12 shows a progressive decrease in the number of pinholes with the increase in coating thickness, with this decrease being more substantial for the thicker coating. Appendix A shows the microscope images of Cr coatings on PCB, using the retro-illumination mode, in order to observe the number of pinholes present on the surface. As it is portrayed in the literature, increasing the deposition time can result in lower defect densities, especially at low deposition temperatures [43].

#### 3.2.5. Hardness and Young’s Modulus

After performing the nanoindentation tests, the films’ hardness and reduced Young’s modulus were calculated for the films deposited over Si substrates, as shown in Figure 13. 

Regarding the reduced Young’s modulus, the obtained values are approximately constant for 800 nm and 1400 nm and then decrease for 1600 nm. For the 800 nm and 1400 nm films, the obtained values were 210.5 GPa and 214.1 GPa, respectively. These values are within the range of values reported in the literature for Cr thin films [44,45]. The lowest observed value was 187.5 GPa, for the 1600 nm film. Such difference may be due to the more pronounced changes in the chromium coating formation that may have caused a slight change in its mechanical properties [22,44]. Lintymer et al. [46] demonstrated that the Young’s modulus of sputtered Cr films decreases with increasing porosity following a perfect mixture rule. Therefore, the films deposited with 1600 nm of thickness should contain a certain degree of porosity, which can be intra-granular and/or inter-columnar.

According to Figure 13, there is a decrease in hardness with the thickness increase from 6.8 GPa (800 nm) to 4.6 GPa (1600 nm). Different factors could influence this variation. First of all, to avoid the influence of the substrate during the indentation test, it is important to set the indentation depth to less than 10% of film thickness [45,47,48]. For the measurements on the Cr films, this factor was taken into account and the indentation depth was set to avoid the influence of the substrate. Additionally, for thin films of similar chemical composition, the hardness strongly depends on two main factors: the residual stresses and the crystal size in the coatings [49]. The former factor can be neglected since the residual stresses in the films deposited over Si were negligible for all coating thicknesses (see Appendix A). Concerning the latter factor, the estimated crystal size (of Cr films deposited over polymer, in Figure 5, which can give a good idea of the expected behavior of Cr films over Si) decreased progressively with the thickness of the films when going from 800 nm to 1600 nm, going below the estimated critical crystal size (*d_c_* ~12 nm) in the Hall–Petch effect. Therefore, the inverse Hall–Petch relationship could help explain the hardness reduction from 800 nm to 1600 nm. According to the inverse Hall–Petch relationship, as the crystal size becomes smaller (for crystal size values below the critical crystal size), larger fractions of atoms belong to the crystal boundaries and crystal boundary sliding becomes easier, resulting in a softening of the material for smaller crystal sizes [49]. The reduction in hardness could also be explained with the presence of inter-columnar voids that might appear more pronouncedly as the columns become bigger and less dense [31].

## 4. Conclusions

Chromium metal films with increasing thicknesses (800 nm, 1400 nm, and 1600 nm) were sputter-deposited on PC polymer substrates. Prior to each deposition, a short plasma etching cleaning procedure was applied.

Combined SEM and AFM analysis confirmed the presence of a columnar growth structure in the Cr films. For the 800 nm and 1400 nm thick films the columns were denser, whereas for the 1600 nm thick films the columns were bigger and more open, resulting in a rougher, more heterogeneous surface. Both surface feature size and RMS roughness increased with the coating thickness for both substrates.

Through XRD analysis it was proved that all films show a preferential orientation according to the (222) plane, whose corresponding peak intensity increases with thickness due to the shadow effect.

Cr thin films without any major defects could be deposited for thicknesses up to 1400 nm, after which the films began to form pronounced buckling, for both substrates. The formation of hillocks at 1600 nm was attributed to the high compressive stresses formed during the cool-down step of the substrates after deposition (thermal residual stresses).

The adhesion test confirmed that the films were highly adhesive up to 1600 nm, with the best classification, showing that these coatings can be implemented for industrial applications.

The 800 nm and 1400 nm thick films showed good-enough brightness and color coordinates to be used in the industry. Moreover, there was a decrease in pinhole defect density with the thickness increase and the hardness of the films was close to 6 GPa for all Cr thicknesses, which is acceptable from an industrial point of view.

All the above analysis confirms that the present strategy of Cr sputter deposition with plasma pre-treatment allows Cr coating thicknesses up to 1400 nm on polycarbonate polymers without compromising overall the quality of the films.

## Figures and Tables

**Figure 1 materials-16-02315-f001:**
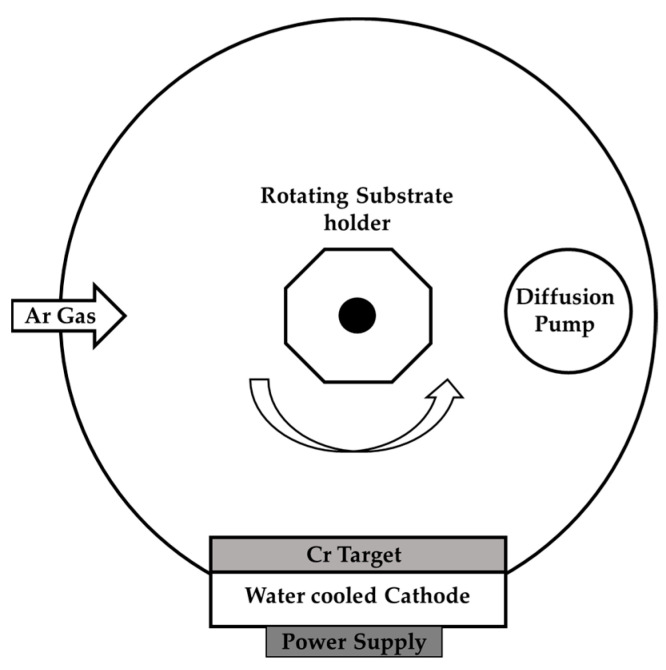
Schematic diagram of the magnetron sputtering deposition chamber.

**Figure 2 materials-16-02315-f002:**
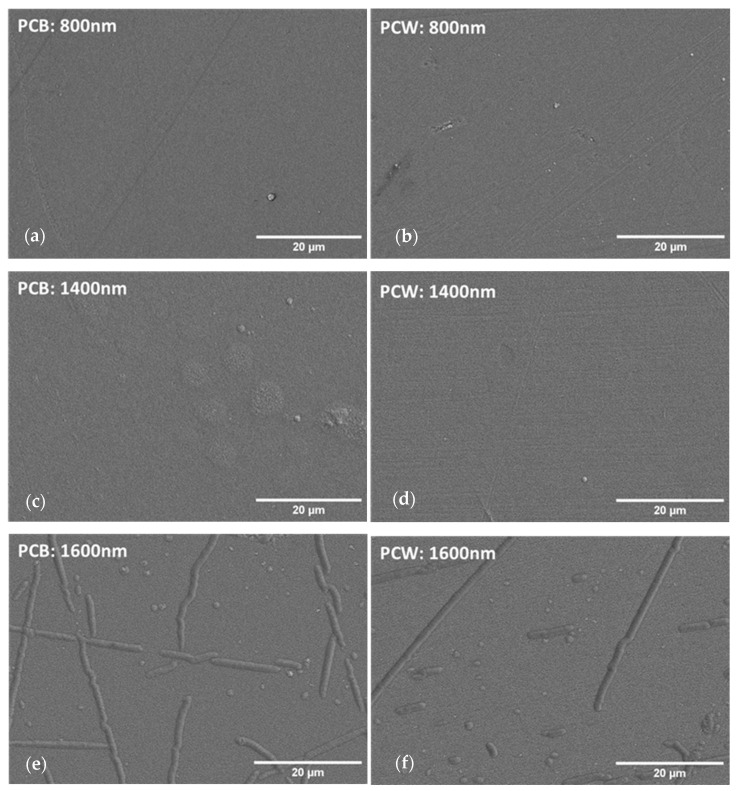
SEM top-view micrographs of coatings with different thicknesses deposited on polymeric substrate: (**a**) 800 nm (PCB), (**b**) 800 nm (PCW) (**c**) 1400 nm (PCB), (**d**) 1400 nm (PCW), (**e**) 1600 nm (PCB), and (**f**) 1600 nm (PCW).

**Figure 3 materials-16-02315-f003:**
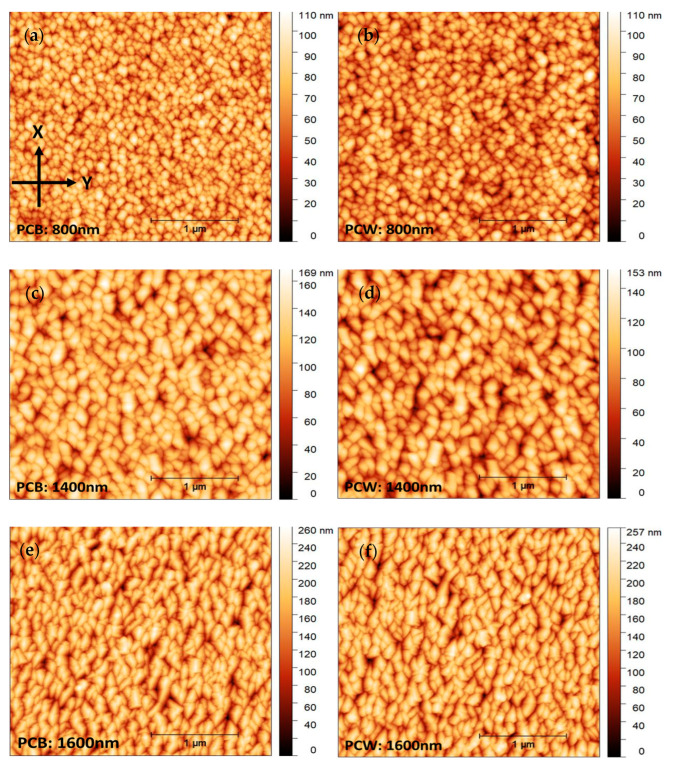
Topographical AFM images of coatings deposited on polymeric substrates: (**a**) PCB (800 nm), (**b**) PCW (800 nm), (**c**) PCB (1400 nm), (**d**) PCW (1400 nm), (**e**) PCB (1600 nm, in an area without hillocks), and (**f**) PCW (1600 nm, in an area without hillocks). The images shown in Figure 4a represent on the Y-axis the tangential component of the substrates’ direction of rotation.

**Figure 4 materials-16-02315-f004:**
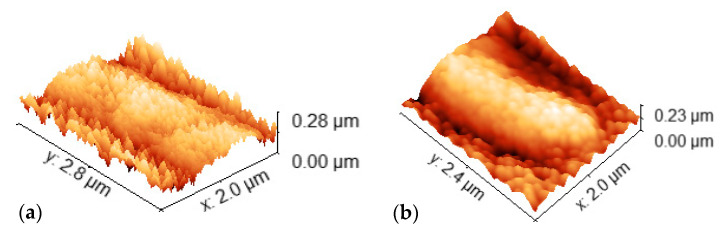
Three-dimensional topographical AFM images of the hillock regions observed in the 1600 nm Cr coatings for: (**a**) PCB and (**b**) PCW.

**Figure 5 materials-16-02315-f005:**
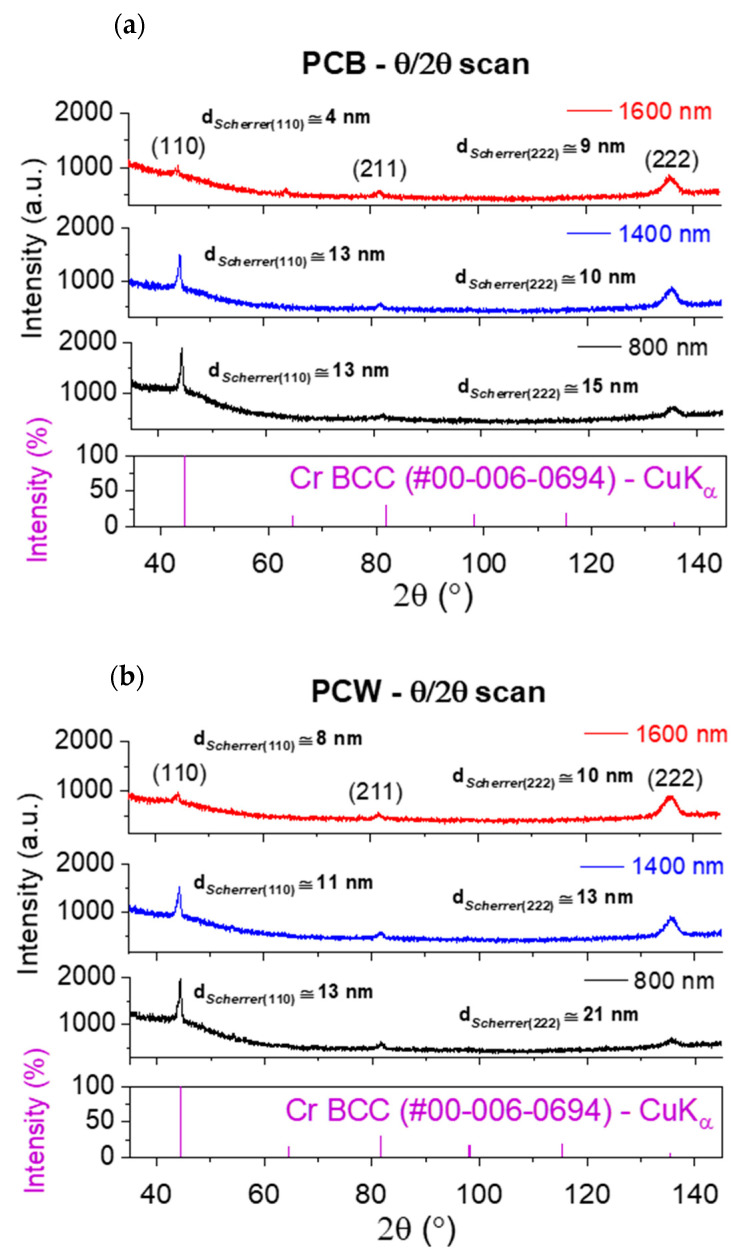
Bragg–Brentano XRD patterns for different thicknesses of Cr films deposited over: (**a**) PCB and (**b**) PCW.

**Figure 6 materials-16-02315-f006:**
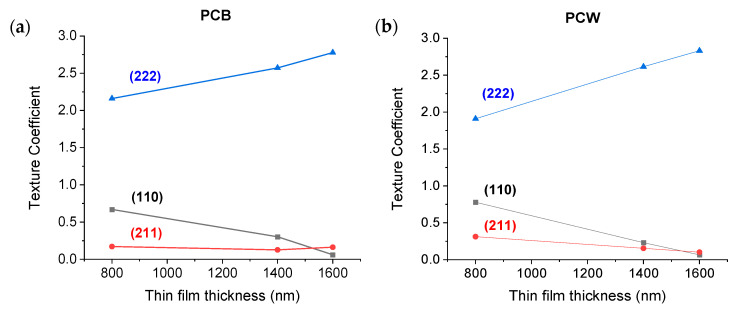
Texture coefficients of Cr films deposited over: (**a**) PCB and (**b**) PCW.

**Figure 7 materials-16-02315-f007:**
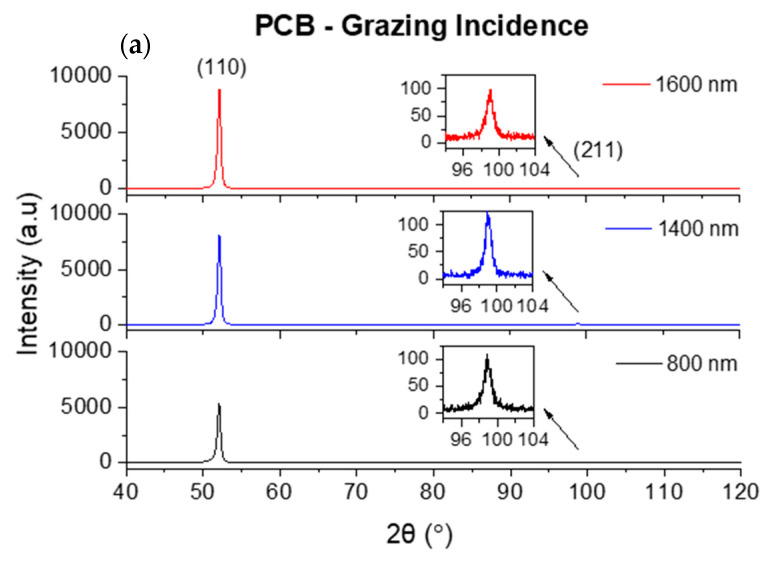
GIXRD patterns for different thicknesses of Cr films deposited over: (**a**) PCB and (**b**) PCW.

**Figure 8 materials-16-02315-f008:**
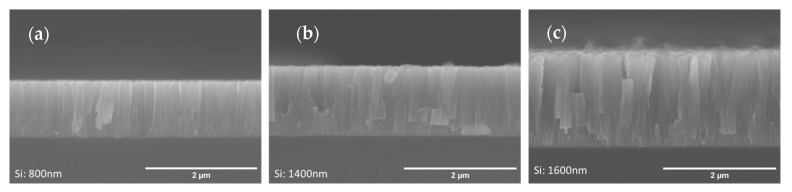
SEM cross-section micrograph images of coatings with different thicknesses deposited on Si: (**a**) 800 nm, (**b**) 1400 nm, and (**c**) 1600 nm.

**Figure 9 materials-16-02315-f009:**
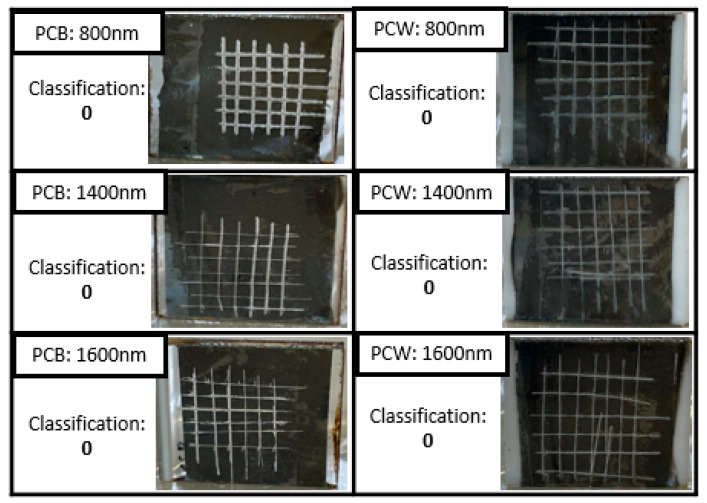
Representative images of the adhesion of the Cr coatings to the substrates, with the respective classification.

**Figure 10 materials-16-02315-f010:**
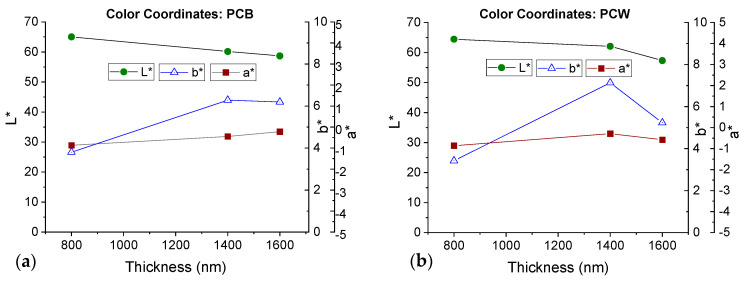
CIE-L*a*b* color coordinates of Cr coatings with various thicknesses for: (**a**) PCB and (**b**) PCW.

**Figure 11 materials-16-02315-f011:**
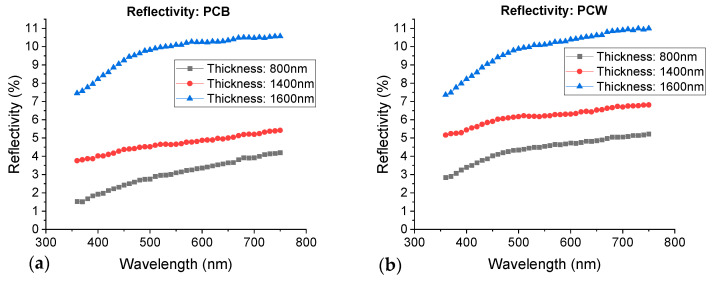
Diffuse reflectivity of Cr coatings deposited over: (**a**) PCB and (**b**) PCW.

**Figure 12 materials-16-02315-f012:**
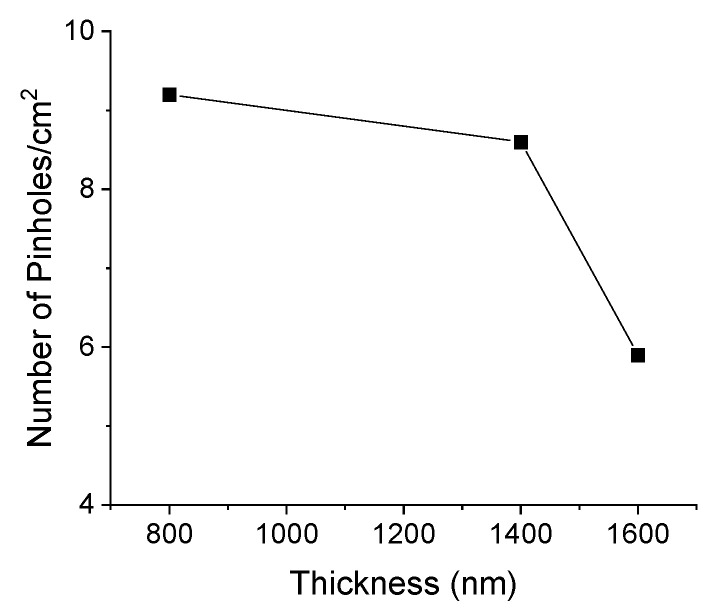
Evolution of the pinholes defect density with the Cr films thickness (PCB).

**Figure 13 materials-16-02315-f013:**
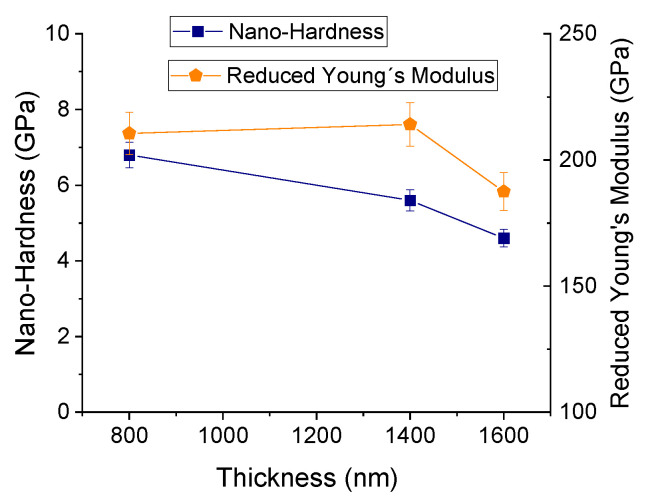
Hardness and reduced modulus of elasticity of the Cr coatings deposited on Si.

**Table 1 materials-16-02315-t001:** Variation of surface feature size and roughness with thickness for PCB and PCW substrates (both parameters were calculated in a buckling-free zone).

Thickness (nm)	Range of the Mean Value (Surface Feature Size (nm))		Roughness (nm)	
	PCB	PCW	PCB	PCW
800	~50–70	~53–70	14	15
1400	~80–110	~70–95	22	22
1600	~130–165	~130–170	35	25

## Data Availability

The data presented in this study are available on request from the corresponding author.

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
