# Peer review of "Decorative Chromium Coatings on Polycarbonate Substrate for the Automotive Industry"

_materials, 2023, doi:10.3390/ma16062315_

Round 1

Reviewer 1 Report

The work evaluates properties of pulsed DCMS chromium coatings on polycarbonate substrates from the viewpoint of the requirements for their applications in automotive industry with the focus on the effect of their thickness. Neither chromium coatings, pulsed DCMS nor methods included in the study are new but because of strong application potential, the work would be worth of publication. However, large number of improvements is required prior to its recommendation for publication. They include:

1. Abstract: the first 5 sentences can be deleted, they are not necessary.

2. Introduction is also very lengthy, especially the second paragraph describes general knowledge available in each handbook on deposition technologies. The introduction should be shortened and rewritten in more straightforward way.

3. Base coat was not described at all, one cannot assess its importance.

4. SI units should be used (Pa instead of mbar).

5. different arrangements of the images in Fig. 1 and Fig.2 is confusing, please, use the same layout.

6. Why buckling occurred along the lines? This is very unusual way of local delamination and it should be at least somehow analyzed.

7. The role of residual stresses is discussed in two different chapters...Is a text rearrangement possible to avoid such duplication? In the field of residual stresses in thin coatings, the works of G. Abadias should be mentioned among the references.

8. Are you sure that the size of the cups (on the top of (each?) columns) obtained from AFM images of coating surfaces corresponds to the size of the grains in the coating? Fig. 4 suggests the presence of columns, can you compare the diameter of these columns with the "grain size"from AFM images?

9. Fig. 4 suggest oriented columnar structure....is it textured? Do you expect any correlation between coating thickness and degree of texture with the increase of coating thickness? In the positive case, what kind of influence on mechanical properties can be expected? Do you think texturing and shadowing may be related?

10. what does it mean that the columns are "more open" (in Fig. 4c)....

11. Although the tape tests showed excellent adhesion in all cases, no differences were detected ... The whole chapter 3.2.1. can be therefore replace just by sentence...Maybe another adhesion tests (e.g. microscratch) would be more appropriate to reveal the differences?

12. The residual stresses measured on Si substrate are, in fact, not relevant to the work at all. This part should be focused on the stresses in the coating on PC, otherwise it is confusing (moreover, the  stresses in the coating on Si are negligibly small).

13. The way nanoindentation data were evaluated is highly questionable. According to ISO 14577-4, hardness and indentation modulus of the coatings have to be evaluated from the plateau on the corresponding depth profile extrapolated to zero depth. Selecting one depth which is within 10% of thickness is definitely not sufficient! Moreover, lower limit of each depth profile suitable for the evaluation is strongly affected by the blunting of the indenter tip; its surface area calibration usually suggests that the minimum depth, when correct values of indentation modulus (and hardness, too) are calculated, is usually close to or even above 100 nm. Without depth profiles, one cannot assure that the values calculated from single depth measurements fall within proper depth range.  Moreover, it is not clear, why mechanical properties were measured only on Si substrate and not also on both PC substrates as expected from the aims of the work.

The above remarks suggest that the work requires major improvements not only in the measurement methods but also in the analysis and understanding of the obtained results, way they are described and discussed.  WIthout these changes the work cannot be recommended for publication.       

Reviewer 2 Report

Please see the enclosed comment file.

Reviewer 3 Report

In this paper, Cr thin films were prepared by magnetron sputtering, and chromium coatings with different thickness were deposited on two different substrates. The effects of film thickness on its structure and properties were studied. The research results have the following problems:

1 In Part 3.1, in the characterization of the microstructure and surface topography of the film, it is not enough for the author to determine the microstructure of the film only through SEM and AFM analysis. The grain size and the growth morphology of the surface section of the film are closely related to the phase and preferred orientation of the film. It is suggested that the author supplement the XRD diffraction analysis of the film and further analyze the grain size and microstructure changes of the film in combination with the diffraction results.

2 How is the residual stress of films with different thickness obtained in Figure 9? Please explain how the thickness of the film affects the stress of the film.

Additionally, for thin films of similar chemical composition, the hardness strongly depends on two main factors: the grain size and the residual stresses of the coatings [41].Only the change and influence of grain size are mentioned in the paper. Please analyze the influence mechanism of stress.

Reviewer 4 Report

Authors of the paper studied decorative chromium coatings deposited on a polycarbonate substrate (PC) which were expected to be used in the automotive industry. The idea of the study is still up to date in view of using the hard chromium for decorative coatings and important thanks to the durability of the final part. The method used – magnetron sputtering onto polycarbonate (PC) substrates – is effective due to the ease of deposition with incremental thickness, on request.

          The method of the study is typical, right and the authors discussed their promising results before conclusion. The Supplementary Materials are useful, and may be left. Two >Figures one< should be corrected. There are some other faults (e.g. „37 C degree” line 101; >> young’s modulus<< beginning from line 151;  or >>10 %<<) which should be corrected in the manuscript. I wonder why there are no Highlights nor Graphical abstract presented in the manuscript, which could be useful.

Round 2

Reviewer 1 Report

The main problem of the manuscript seems to be the novelty. Chromium  is possibly the most common bond layer applied to many hard coatings deposited by magnetron  sputtering. Maximum one sentence mentioning it is devoted to it in the experimental part in these papers because...no problems. The current work just confirms it and its main result (at least for me) is that DCMS Cr coatings are without problems up to around 1.4 um, thicker coatings would delaminate. Is this information sufficiently new and important to be published in an international scientific journal?  I doubt it and from my viewpoint, it is only of technological importance for the producer and for a given application.

Moreover, the manuscript can be improved:

1. the role of the residual stresses is discussed in confusing way: Fig. S2 shows that the compressive stresses were around and below 0.2 GPa which means they are negligibly small . On the other hand, thermal stresses were discussed at least twice despite neither a difference in temperature nor in thermal expansion coefficients are sufficient to generate them (which agrees with low residual stresses). Then, the reason for buckling in the coatings with 1600 nm thickness cannot be explained by (negligible) residual stresses. 

2. the idea of shadow effects in the formation of 3-sided pyramidal tops seems to be highly speculative. The evidence for such geometry on AFM images is not sufficient, the geometry must be shown very clearly to justify such claim. Moreover, rotation applied in your case would result in the same shadowing in rotation direction and mostly eliminate it.

3. Grain/column coarsening can be expected from Tab. 1 and fracture surfaces (Fig. 8) whereas grain refinement indicated by XRD was used in the discussion. Apparently, the controversy may arise due to the difference between grains and crystallites (within these grains). It is supported also by the difference between the diameter of the columns (in 100 nm range in Fig. 8c) while only 9-10 nm was obtained from Scherrer equation.

4. the comparison of  "compactness" (term "density" would be more appropriate) based on low magnification images of fracture surfaces seems to be speculative, better evidence is necessary.

5. The end of the discussion about hardness and indentation modulus degradation at grater thicknesses is confusing. Hall-Petch relations was discussed as the main reason and then it was completely undermined by the last sentence suggesting the reduction of hardness could be due to porosity (called intercolumnar voids).

5. English improvements: spell checking would eliminate small mistakes like: mom solution (p.2, r.75), Regaku instead of Rigaku (p.3, r. 128), caption in Fig. 1 in separate row.

The improvements addressing the above remarks could improve the manuscript but the problems with the novelty and reason to publish it in an international scientific journal persist. Because of that I doubt it is suitable for the publication in Materials.

Reviewer 3 Report

The author has supplemented the data and now it is acceptable.

Author Response

The reviewer has considered the article acceptable.